# Beyond Typical Electrolytes for Energy Dense Batteries

**DOI:** 10.3390/molecules25081791

**Published:** 2020-04-14

**Authors:** Rana Mohtadi

**Affiliations:** Materials Research Department, Toyota Research Institute of North America, Ann Arbor, MI 48105, USA; rana.mohtadi@toyota.com; Tel.: +1-734-995-4012

**Keywords:** electrolyte, hydride, battery

## Abstract

The ever-rising demands for energy dense electrochemical storage systems have been driving interests in beyond Li-ion batteries such as those based on lithium and magnesium metals. These high energy density batteries suffer from several challenges, several of which stem from the flammability/volatility of the electrolytes and/or instability of the electrolytes with either the negative, positive electrode or both. Recently, hydride-based electrolytes have been paving the way towards overcoming these issues. Namely, highly performing solid-state electrolytes have been reported and several key challenges in multivalent batteries were overcome. In this review, the classes of hydride-based electrolytes reported for energy dense batteries are discussed. Future perspectives are presented to guide research directions in this field.

## 1. Introduction

Over the past few decades, nickel-metal hydride and later lithium-ion batteries, have been thrusted to the forefront of societal electrification owing to their energy density and reliable performances. For instance, the relatively high specific energy densities of Li-ion batteries (*ca* at least 2 times that in nickel-metal hydride) and higher specific power have positioned them to play an increasingly important role in driving technologies encompassing miniature and portable devices (i.e., cell phones, laptops), medium scale (plug-in hybrid PHEV and electric vehicles EVs) up to large scale stationary and grid applications [1].However, recently there have been increasing demands for highly performing batteries beyond those based on typical Li-ion [2]. In particular, the energy and power densities that can be offered by Li-ion batteries are insufficient to meet those needs. This has been fueling R&D efforts towards battery chemistries capable of meeting these requirements that include solid-state batteries and those based on high capacity metals such as Li and Mg. For example, in contrast to batteries utilizing liquid electrolytes, the use of solid-state electrolytes allows for efficient bipolar stacking design of batteries that decreases the dead space between single cells thereby increasing the overall energy density whilst eliminating the use of volatile liquid electrolytes [3,4]. In addition, the high hardness of solid electrolytes was suggested to mitigate the occurrence of Li dendrites that form upon Li metal battery charge/discharge. On the other hand, batteries that use multivalent metals have been promising a high energy density option that uses benign and earth abundant metals such as Mg (volumetric storage capacity is 3832 mAh/cm^3^ vs. 2061 mAh/cm^3^ for Li metal) and eliminates safety hazards encountered with Li metal.

Enabling batteries beyond Li-ion mandates the presence of highly performing battery core components (anode, cathode, electrolyte) capable of supporting competitive battery performances. Electrolytes, which are the media that connect the positive (cathode) and negative (anode) electrodes, are positioned to play a critical role in enabling these technologies. However, whilst numerous families of materials have been proposed for these batteries, they fall short of simultaneously meeting the challenging technological needs. For example, highly conducting Li^+^ solid-state electrolytes such as sulfides suffer from limitations in their electrochemical stability window [3,5]. In addition, electrolytes used in multivalent batteries had severe limitations caused by their reliance on complex chloro-reagents or were altogether incapable of supporting highly efficient performances [2,6,7] Therefore, there have been dedicated efforts that target the development of new materials positioned to overcome these challenges.

The past decade has witnessed unexpected and rapid developments of hydrides-type materials as competent solid and liquid electrolytes poised to offer potential solutions [8]. This review explains how these advancements came to fruition, summarizes the classes of hydride-based electrolytes and highlights key advancements made thus far. 

## 2. Explanation of the Review’s Structure 

The development of practical electrolytes for energy storage devices is a very challenging endeavor as they are required to meet a myriad of requirements (Figure 1) [9]. All of these demands are based on a battery cell design that allows for optimized battery performances and offer a commercially viable energy storage system. For example, high conductivity is required to enable acceptable battery discharge/charge rate capabilities; wide electrochemical window is needed to prevent electrolyte decomposition that can result in battery failure (i.e., electrolyte is compatible with battery electrodes); absence of no-self discharge or electronically insulating electrolyte is required to prevent battery capacity loss whilst low/non-volatility is highly desired to minimize safety risks and lower manufacturing costs. This review explains how hydride electrolytes for mono and multivalent batteries have been shown to meet several of these requirements which included the demonstration of high conductivity, wide electrochemical window and non/low volatility.

Hydrides are materials characterized by the presence of a hydrogen atom bound with many elements forming ionic (i.e., negatively charged H^−^ or protonic H^+^ form), covalent, or interstitial systems [8]. In particular, those constituting of anions formed by H^−^ covalently bonding to a metal such as boron have been attracting tremendous attention over the past few years as battery electrolytes. The high potential of hydrides as electrolytes was first reported in the borohydride salts and was manifested by two key accomplishments [8]: (1) The discovery of high Li^+^ ion conductivity (*ca* 10^−3^ S/cm at 423K) and wide electrochemical window (5V) in the solid state (discussed in Section 3) and (2) The design of boron-hydrogen Mg battery liquid electrolytes that are highly compatible with the passivation prone Mg metal (>90% Mg deposition/stripping efficiency) (discussed in Section 4). A timeline depicting key developments is captured in Figure 1b.

The research of hydrides as solid-state electrolytes for monovalent batteries has been motivated by improving the room temperature conductivities of Li^+^ and Na^+^ ions (Section 3). High room temperature conductivities in the order of 10^−3^ S/cm, which placed hydride solid-state electrolytes amongst the best-known electrolytes, was the culmination of these efforts. These studies are discussed in Section 3.1, which is dedicated to strongly hydridic salts, and Section 3.2, which discusses electrolyte salts with a weak hydridic nature. Section 4 explains the advancements made in the field of hydrides as electrolytes for multivalent batteries wherein higher than 99% Mg deposition/stripping coulombic efficiency (>90% for Ca deposition/stripping), coupled with a wide electrochemical stability window exceeding 5 V for Mg salts, resulted in major breakthroughs in this field. Section 4.1 discusses the research of hydride as liquid electrolytes for Mg and Ca batteries that was driven by enabling a high compatibility with these metals. These studies unveiled the limitations of liquid electrolytes employing salts with a highly hydritic nature as was manifested by the low anodic stability of the borohydride salts (i.e., <2 V vs. Mg/Mg^2+^). Section 4.2 discusses the design concepts that improved the anodic stability of hydrides whilst maintaining the compatibility with Mg metal accomplished using *closo*-borate salts. The findings reported in Section 4.1 and Section 4.2 most recently paved the way for important Mg metal-hydride fundamental interfacial studies (Section 4.3) and the design of solid-state Mg electrolytes (Section 4.4). Section 5 discusses a very recent new application of hydrides as non-volatile and highly stable battery solvents (ionic liquids), which are amongst the best known ionic liquid electrolytes due to their wide electrochemical window (>4 V) and compatibility with both monovalent and multivalent metals. Finally, Section 6 showcases key battery demonstrations of hydrides as electrolytes in mono and multivalent batteries.

## 3. Solid-State Hydride Electrolytes for Monovalent Batteries

The observation of appreciable conductivity at room temperature in a hydride was first reported close to four decades ago in Li_2_NH (ca 10^−4^ S/cm, ca 10^−2^ S/cm at 400 K) [10]. The ionic conduction was mediated by Frenkel pair defects and/or charged vacancies and the cation diffusion occurred via pathways, which involved octahedral to tetrahedral jumps along the (001) crystallographic direction. However, its poor electrochemical stability (0.7 V vs. Li^+^/Li) and high reactivity made it less attractive amongst other solid-state electrolytes researched at that time. This interest in the solid-state cationic conductivity in hydrides was revived from 2007 with the unexpected discovery of high Li^+^ mobility after microwave hearting of LiBH_4_ [11]. This salt has been known to undergo phase transition from orthohombic to hexagonal phases at 380 K [12], however, the mobility of Li^+^ was revealed from permittivity measurements that demonstrated the presence of conductive heat loss in the hexagonal phase which suggested Li^+^ ion mobility (hydrogen is immobile as it is covalently bonded in BH_4_^−^). Electrochemical impedance measurements (Figure 2) further confirmed the presence of high Li^+^ conductivity in the hexagonal high temperature phase (in the order of 10^−3^ S/cm). In this phase, the high mobility of Li^+^ ions was explained by the presence of rotational disorder in the BH_4_^−^, the formation of metastable interstitial sites and the alignment of both the Li^+^ and BH_4_^−^ along the *a* and *b* axes thereby permitting unobstructed migration of Li^+^ in both directions [13,14,15]. The salt was shown to have high oxidative stability beyond 5 V vs. Li^+^/Li, however this stability was later shown to be apparent and not representative of the thermodynamic stability of this salt. For example, the anodic stability was measured to be about 2 V vs. Li/Li^+^, when large surface area carbon was infused within the electrolyte [16]. The seminal discovery of Li^+^ high mobility in the solid state triggered research interests to investigate hydrides as solid-state electrolytes for Li^+^ and Na^+^ ion batteries. These are described in the next Sub-Sections. 

### 3.1. Development of Highly Conductive Solid Electrolytes

In an effort to lower the phase transition temperature of LiBH_4_ to a superconductor, i.e., enable high conductivity at room temperature, the concept of stabilizing the hexagonal phase through the use of halide dopants (Br^−^, Cl^−^ and I^−^), used to partially substitute the BH_4_^−^ anion through forming solid solutions, was conceived and investigated [17,18,19]. This approach was proven most successful for (1 − *x*)Li(BH_4_) + *x*LiI solid solution, where a hexagonal crystal at room temperature (*x* > 13%) could be formed using mechanochemical synthesis and demonstrated high Li^+^ conductivities, i.e., 0.1 × 10^−3^ S/cm at 303 K for the solid solution with 40% LiI. 

Beyond the aforementioned concept, other studies increased the room temperature ionic conductivities through the formation of mixed borohydride-amide salts as was shown for Li^+^ in trigonal Li_2_(BH_4_)(NH_2_) and cubic Li_4_(BH_4_)(NH_2_)_3_ where high conductivities in the order of 10^−4^ S/cm were achieved at room temperature [20]. The key for enabling these high conductivities was attributed to the presence of multiple Li^+^ crystallographic sites [20]. Interestingly, the activation energy for Li_4_(BH_4_)(NH_2_)_3_ (0.26 eV) was lower than the hexagonal superconductor LiBH_4_ phase(0.53 eV) [13,20]. It is worth noting that Li_2_(BH_4_)(NH_2_) became a molten salt at a relatively low temperature (onset *ca* 350 K). Further increase of the LiBH_4_/LiNH_2_ to 1:2 resulted in high conductivities (6.4 × 10^−3^ S/cm at 313 K) through the formation of cubic Li_3_(BH_4_)(NH_2_)_2_ phase (isostructural to Li_4_(BH_4_)(NH_2_)_3_) [21]. This salt underwent partial melting at 313 K, so it would be insightful to determine the contribution of the liquid phase to the overall bulk conductivity.

Other borohydride salts studied that increased the conductivity of Li^+^ included mixed-cation mixed-anion borohydrides [22,23]. In particular, in LiM(BH_4_)_3_Cl (M = cation), the disordered distribution of Li^+^ increased the ionic conductivity at 293 K to 1.02 × 10^−4^, 3.5 × 10^−4^, 2.3 × 10^−4^ S/cm for Ce, Gd and La-containing compounds, respectively. In addition, bimetallic borohydride oxide [24] i.e., LiCa_3_(BH_4_)(BO_3_)_2_ had a Li^+^ conductivity of 2.5 × 10^−6^ S/cm at room temperature which increased by about an order of magnitude when prepared with excess lithium and doped with either heterovalent Na^+^ or homovalent Sr^2+^(conductivities in the order of 10^−5^ S/cm at room temperature) [25]. A structural investigation of LiCa_3_(BH_4_)(BO_3_)_2_ showed that the borohydride was not directly involved in the conduction mechanism as (calcium borohydride substructure served to stabilize the percolating pathway) the conduction paths were composed exclusively of BO_3_^3−^ anions, which formed faced connected tetra- and octahedra accessible sites for Li^+^ jumps and suggested the vacancy-dependent mobility of Li^+^.

Alternate approaches that sought to increase the conductivity of LiBH_4_ utilized composites which included those with nano oxides or sulfides. In the former case, high room temperature conductivity (10^−4^ S/cm) was observed for the LiBH_4_–SiO_2_ nanocomposite prepared by melt infiltration of the borohydride into a mesoporous inorganic silica scaffold (Figure 3) [26,27]. The study suggested that the high mobility of Li^+^ was not related to stabilization of the hexagonal phase but was presumably caused by the high density of defects and low diffusion barriers at the interface between the two solids, which resulted from disorder, strain and space-charge regions [27,28]. Follow up studies demonstrated a similar effect using Al_2_O_3_ [29] and C_60_ [30] additives (note that the electronic conductivity in the C_60_ [30] composites could be higher than that desired from a solid electrolyte). The formation of composites with sulfides was first reported for LiBH_4_:P_2_S_5_, where the preparation process resulted in a material that went beyond a composite formation evident from a new crystalline 90LiBH_4_:10P_2_S_5_ phase that exhibited a high conductivity in the order of 10^−3^ S cm^−1^ around 300 K [31]. The combination of LiBH_4_ with glassy sulfide glass electrolytes *x*LiBH_4_–(100 − *x*):0.75Li_2_S:0.25P_2_S_5_ [32] using ball milling resulted in improved conductivity (i.e., at *x* = 33 it showed 1.6 × 10^−3^ S/cm) and low activation energy (0.30 eV). Interestingly, Raman spectroscopy suggested that BH_4_^−^ was in the same state as that occurring in the hexagonal superconducting LiBH_4_ phase, where high rotational freedom and delocalized negative charge weakened the electrostatic interactions between Li^+^ and BH_4_^−^. The extension of composite studies to Li(BH_4_)_3_I resulted in similar improvements [33]. Other composites that yielded varying degrees of conductivity improvements, albeit less than those achieved with the sulfides, included composites with other borohydrides as Ca(BH_4_)_2_ [34], NaBH_4_ [35], with hydrides such as MgH_2_ [36], or borohydride-hydride ternary mixtures [37], and mixtures with halides such as NaCl [38]. 

Neutral molecules that are adducted to LiBH_4_ were also investigated to increase the room temperature conductivity, as was demonstrated for solid-state lithium borohydride ammoniates Li(NH_3_)_x_BH_4_, x = 0.5 and 1.0 [39]. A dramatic change in the ionic conductivity was attributed to the formation of defects induced by the partial desorption of NH_3_. For example, for Li(NH_3_)BH_4_, the conductivity increased from 1.5 ×10^−6^ S/cm at 303 K to 2.21 x 10^−3^ S/cm at temperatures > 313 K. Note that a closed system is required for these electrolytes so that the equilibrium pressure of ammonia is achieved, otherwise, reformation of LiBH_4_ will occur. Detailed structural analyses are desired to establish the conduction mechanism in these salts.

Unlike the case for the lithium salts, the formation of borohydrides that are highly Na^+^ conductive was less successful [40,41,42]. For example at room temperature, the conductivity of Na^+^ in NaBH_4_, NaNH_2_, and Na_3_(BH_4_)_2_I were in the range 10^−10^–10^−9^ S/cm. Disordered distribution of Na^+^ resulted in increased mobility, as was reported for Na_2_(BH_4_)(NH_2_) (ca 10^−6^ S/cm at 300 K), however the conductivity remained inferior to that reported in the lithium salts [40,41].

Inspired by the promising results in borohydrides, other complex hydrides such as alanates and amides were investigated [41,43]. In these cases, no transition to a super conducting phase was observed in the alanates leading to relatively low cationic mobilities [43]. Likewise, with the exception of Li_2_NH, imides and amides were less successful in demonstrating a good potential as solid-state electrolytes [44,45,46]. 

### 3.2. Towards Achieving High Li^+^, Na^+^ Mobilities: Closo-Borates 

Investigations of ionic conduction mechanisms in borohydrides showed that it is not only vacancy-dependent but also based on the “paddle-wheel” mechanism [47,48,49] wherein the high rotational mobility of the (M_y_X_n_)^δ−^ units promotes the ionic conductivity and lowers the activation energy [50,51]. This concept inspired investigating boron clusters (*closo*-borates)-based salts as electrolytes. These are found as undesirable byproducts that form during the thermal dehydrogenation of borohydrides [52]. Namely, the structural evolution of Li_2_B_12_H_12_ with temperature indicated possible Li^+^ conduction following transformation at 638 K to a disordered phase (β phase) with a frustrated Li^+^ lattice [53]. In addition, a solid-state nuclear magnetic resonance (NMR) study of the spin-lattice relaxation in Na_2_B_12_H_12_ salts revealed a substantial increase in the reorientational jump rates of B_12_H_12_^2−^ accompanied by the fast translational diffusion of Na^+^ following a first-order transition near 520 K [54]. Interestingly, investigating *closo*-borates as solid electrolytes for Li and Na batteries was reported concurrent with the report of Mg *closo*-borate salts as competent liquid electrolytes for Mg batteries- owing to their compatibility with Mg metal anode and high anodic stability- as discussed in Section 4 [9].

High cationic translational mobility in boron clusters was first reported for hydroborates in Na_2_B_12_H_12_ where high Na^+^ conductivity in the order of 10^−1^ S/cm, near 540 K was obtained (Figure 3) [55]. These conductivities were observed following transition from a low-temperature (ordered monoclinic) to a high-temperature (disordered, body-centered cubic) phase and were consistent with ^23^Na NMR measurements that showed enhancements in Na^+^ cation jump rate (>2 × 10^8^ jumps/s) in the high temperature polymorph [54]. Lowering the transition temperature to a super conductor was reported for Na_2_B_10_H_10_, where a disordered face-centered cubic phase (>360 K), allowed for high Na conductivities in the order of 0.01 S/cm at 383 K [56]. One source of this improvement was speculated to result from the less spherical B_10_H_10_^2−^ anion which offered improved cationic diffusion. Results suggested that the anion dynamics was one major contributors to the conduction of the cation. For example, ab initio molecular dynamics (AIMD) simulations showed that the Li^+^ ion diffusivity in β-Li_2_B_12_H_12_ would be reduced by three orders of magnitude if the anion was constrained and immobile [57]. Computational studies however also showed that anionic dynamics (rotation/vibrations), is not the only factor that dictates the cationic mobility, rather, it is a complex interplay between the density of accessible diffusion sites, anionic dynamics and the nature of local bonding [58]. 

The relatively high mobility of Na^+^ in *closo-*borates triggered further studies to further improve the room temperature ionic conductivity. Similar approaches to those proven successful in enhancing the conductivity of Li^+^ in LiBH_4_ were implemented such as partial substitution with another anion including BH_4_^−^ and a halide (Br, Cl, I) or with a cation like Li^+^ [59,60]. The absence of order–disorder transition and high room temperature conductivity (0.5 × 10^−3^ S/cm) for Na_3_BH_4_B_12_H_12_ suggested that the anion dynamics were a less important contributor to the superconductivities and that structural factors were more at play in this case. Cationic substitution was reported to enhance the ionic conductivity as in LiNaB_12_H_12_ (i.e., at 550K, the conductivity was eight times higher than Na_2_B_12_H_12_). Both Li^+^ and Na^+^ cations were mobile; Na^+^ became more mobile between 393 K to 433 K as suggested by the decreased Li^+^ transference number, respectively, from 0.91 to 0.71 [60]. However, these improvements in conductivity were later suggested to result from the inadvertent presence of other cluster fragments, i.e., B_10_H_10_^2−^, produced from the synthesis of the salts using B_10_H_14_. For example, intentional inclusion of B_10_H_10_^2−^ in the structure of Na_2_B_12_H_12_ or simply preparing a binary complex of these anions dramatically enhanced the cationic conductivity [61,62]. It is noteworthy to mention that similar effects of these fragments on improving the cationic conductivity of Li_2_B_12_H_12_ (10^−4^ S/cm), prepared from B_10_H_14_ [63]; believed to result from a brief ball milling, were also suggested [64]. Systematic studies of possible extended ball milling effects on conductivity improvement in several boron cluster salts free from other undetermined cluster fragments were later reported, aided by powder x-ray diffraction (PXRD) and Quasielastic Neutron Scattering (QENS). The main finding was that ball milling resulted in the presence of the high temperature disordered phase at room temperature in the processed materials, which suggested a stabilized room temperature disordered form [65]. Conductivity enhancements were drastic, i.e., at room temperature the conductivity for Na_2_B_12_H_12_ was three orders of magnitude higher than that of the pristine sample. It is interesting that these ball-milled electrolytes consisted of both the low conducting and superconducting phases, where the latter phase was present in a form of interconnected nanocrystallites that were distributed in larger crystallites, which exhibited typical bulk-like conductivities [65]. Revisiting the ball milling effects on the conductivity of fragment-free Li_2_B_12_H_12_ (measured in the order of 10^−5^ S/cm at room temperature) was conducted in a later study. This work revealed that the improvements were due to Li^+^ and H deficiencies and proposed that these enhancements were unlikely to result from a stabilization effect of the high temperature phase as was evident from structural analysis. [66] What stands out in this study is the finding that deficiencies of H in boron clusters, that are stabilized by the cation, could be utilized to improve cationic mobilities.

To further improve the cationic conductivity, the utilization of carborate boron clusters, which carry a mononvalent negative charge, was investigated for LiCB_11_H_12_ and NaCB_11_H_12_ [67]. Interestingly, LiCB_11_H_12_ and NaCB_11_H_12_ underwent transition to a superconducting (>0.1 S/cm) disordered phase at much lower temperatures, 400 K and 380 K, respectively (Figure 3), accompanied by a high rate of anion reorientational jumps (10^10^–10^11^ jumps/s). The cluster’s monovalent charge, presence of less neighbors (cation:anion molar ratio = 1:1), and increased lattice constant were hypothesized to cause these enhanced conductivities. These were confirmed from ab initio molecular dynamics (AIMD) which also demonstrated that formation of a dipole (carbon atoms), creates a frustrated lattice and counteracts the ability of the phase to order, thereby reducing the transition temperature to superconducting phases [68].

Inspired from improvements in conductivity observed using the smaller B_10_H_10_^2−^, studies examined Li and NaCB_9_H_10_ salts and demonstrated impressive room temperature conductivities of about 0.03 S/cm in the Na salt (Figure 3). Achieving this conductivity required preheating (i.e., sample conditioning to access the superconducting phase) to about 425 K [69]. Note that systematic studies on the effect of thermal cycling on the room temperature stability of the disordered superconducting phase are needed to discern the origin of these drastic improvements, especially given the hysteresis observed. Substantial improvements were also found for the Li-based salt (Figure 3), i.e., 0.03 S/cm at 354 K. As the case in NaCB_11_H_12_ and LiCB_11_H_12_ salts, the dipole produced by the carbon atom could influence the anion orientations and create a frustrated landscape that results in high cationic mobility at lower temperatures.

Further studies with Na and Li carborate salts targeted stabilization studies of the high temperature superconducting phase and were inspired by those implemented in borohydrides and hydroborates. Stabilization of the disordered phase was studied through formation of what was suggested as mixed (CB_9_H_10_:CB_11_H_12_)^2−^ Li and Na salts [64,65]. These salts were produced from mechanochemical treatment or simple mixing of (CB_9_H_10_:CB_11_H_12_)^2−^ Li and Na salts in the 1:1 molar ratio in aqueous solutions. The absence of phase transitions suggested some sort of stabilization effect on the disordered phases. High ionic conductivity of Li^+^ was found (10^−3^ S/cm at 300 K), and impressive Na^+^ conductivity was reported (6x10^−2^ S/cm at 300 K). A recent report examined the Li salt conductivity in mixed 7:3 molar CB_9_H_10_^−^:CB_11_H_12_^−^ prepared by ball milling also demonstrated very high conductivity of 6.7 × 10^−3^ S/cm at room temperature without thermal activation (Figure 4) [70]. Other studies examined mixing Na carborate CB_11_H_12_^−^ with hydroborate B_12_H_12_^−^ salts and also showed high conductivities (2 × 10^−3^ S/cm at room temperature), albeit lower than those observed for *closo*-carborates [71]. Studies in carborate-type derivatives based on several *nido*-carborates demonstrated inferior conductivities to those reported in *closo-*carborates [72] (note: surprisingly a high value in the order of 10^−3^ S/cm at 300 K was reported for what was described as α-NaCB_11_H_14_, it remains unclear why the conductivity was much higher than that in typical NaCB_11_H_14_, 10^−6^ S/cm at 300 K). Although, the electrochemical stabilities of these specific salts were not reported, they are expected to exhibit a much narrower window compared to the *closo*-borates owing to the known limited stability of the *nido*-anions [72]. 

One approach sought to increase the conductivity in B_12_H_12_^2−^ by confinement in a nanoporous silica SBA scaffold, which was inspired from LiBH_4_ studies discussed above [73]. However, nanoconfinement in the case of Li_2_B_12_H_12_ was not effective as was evident from a very low conductivity compared to that of the bulk material (i.e., 10^−7^ S/cm at room temperature). 

It is worth noting there have been studies that investigated the cationic conductivities in boron clusters that were halogenated. Replacing the hydrogens in the B_12_H_12_^2−^ anion with halogens in the Na salt was counterproductive and showed dramatically low conductivities at room temperature, where high conductivities in the order of 10^−1^ S/cm, could only be achieved at temperatures exceeding 673 K. This behavior was suggested to result from restriction in cluster’s reorientational mobility, caused by increased anionic mass wherein strong bonds to Na^+^ as a result of directional charge distribution on the halogen atoms, contributed to this behavior [74]. A recent computational study supported this finding and suggested that partial halogenation may somewhat reduce conductivity losses in halogenated B_12_H_12_^2−^, however the conductivity will be lower than the hydrogenated form [75].

## 4. Electrolytes for Multivalent Batteries

Recently, there have been increased interests in post lithium batteries that include multivalent storage cations such as Al^3+^, Ca^2+^ or Mg^2+^ [2]. These elements are abundant and offer possibilities for battery cost reductions. In addition, their use in the form of metallic anodes offer a safer alternative to reactive Li metal. In particular, magnesium batteries attracted lots of interest due to their high energy density potential. The Mg metal redox potential is low (−2.4 V vs. SHE) and it has high volumetric capacity (3832 mAh/cm vs. 2062 mAh/cm and 1136 mAh/cm, for Li and Na, respectively) [6]. One key hurdle in these batteries is the passivation of Mg metal, due to electrolyte decomposition, in most common salts/solvents in addition to the sluggish diffusion kinetics of Mg^2+^ in solid-state host structures likely caused by its divalent charge [76,77].

Until recently, the Mg passivation challenge has limited the choice of electrolytes to a handful of chloride base salts/reagents as they were not shown to passivate Mg metal [2,6]. These electrolytes, while highly performing, are corrosive, have limited anodic stability (i.e., limited by Cl^−^ redox potential) and negatively interfere with cation insertion in the cathode [78]. Although designing other alternatives that overcome these challenges seemed farfetched, boron hydrogen compounds were discovered to offer highly competent and practical Mg electrolytes. As similar passivation issues that occur on Mg metal are encountered for Ca metal, strategies that were successful in Mg batteries were shown as applicable to Ca batteries.

### 4.1. Liquid Borohydride Electrolytes: Achieving Compatibility with Mg Metal 

The notion that simple ionic Mg salts could not function in Mg batteries was altered in 2012, where Mg(BH_4_)_2_ was demonstrated as a first example of halogen-free, simple-type ionic electrolyte in Mg-based batteries [79]. The heart of this concept was that the high reductive stability of Mg(BH_4_)_2_ (i.e., reducing agent) would allow it to withstand the low potential environment of Mg metal anode thereby preventing Mg metal passivation. Ethereal solutions of Mg(BH_4_)_2_ in tetrahydrofuran (THF) solvent enabled reversible Mg plating/stripping. This proof of concept sparked research efforts towards understanding the scarcely explored aprotic solution chemistry of hydrides in order to improve the performance of these electrolytes by determining key factors that govern their electrochemical behavior. Spectroscopic analyses (FTIR, NMR) revealed that the electrochemical performance was dictated by poor dissociation in low dielectric solvents such as THF (owing to the relatively ionic nature of the salt). This was overcome using ethereal solvents with more electron donating oxygen sites such as 1,2 dimethoxyethane (DME) and the addition of additives that further dissociate Mg(BH_4_)_2_ such as LiBH_4_, through potential complexation of the acidic cation with the BH_4_^−^. The combination of the aforementioned strategies resulted in a competent performance (Figure 5) with high current densities (25 mA/cm^2^ stripping peak current), low deposition (−0.3 V)/stripping (0 V) overpotentials and excellent Mg deposition/stripping coulombic efficiency (94%). These findings demonstrated that simple ionic salts could be made compatible with the magnesium metal if the anion in the salt has sufficient reductive stability, thereby creating a new design space of highly performing electrolytes for Mg batteries. The electrochemical performance was found to be governed by the extent of salt dissociation per spectroscopic analyses which revealed the presence of strong association in these electrolytes. The amount of the complex cation MgBH_4_^+^ (i.e., charge carrier) tracked with improved electrochemical performance. Follow up studies examined other chelating solvents such diglyme (DGM) and tetraglyme, which further demonstrated the direct relationship between the number of electrodonating oxygen sites and the coulombic efficiency of Mg plating/stripping [80,81]. In addition, computations were used to understand the role of coordination and the effects of the solvent on the solubility and dissociation in these electrolytes [82,83]. Significant and irreversible salt agglomeration in all glymes ranging from THF to tetraglyme was found in all non-dilute borohydride salt solutions [82]. The agglomeration rate and diffusivity of Mg^2+^ in longer chain ethers such as tetraglyme were at their lowest and tracked with the solvent’s self-diffusivity. 

The report of Mg(BH_4_)_2_ as a competent electrolyte sparked interests in revisiting other ionic salts such as magnesium bis(trifluoromethanesulfonyl)imide Mg(TFSI)_2_, known to passivate Mg metal [84,85]. Similar solution chemistry dominated by Mg association with the anion was found in Mg(TFSI)_2_ electrolytes [86], which underscored the importance of the high reductive stability of the BH_4_^−^ anion in preventing passivation of Mg metal. 

Mg(BH_4_)_2_ was applied to mitigate the passivation effects that take place in the presence of Mg passivating anions, i.e., TFSI, present in inorganic salts (Mg(TFSI)_2_) [87], in organic salts (ionic liquids solvents) [88,89] and as a reagent in complex electrolyte solutions such as alkoxyborates (i.e., mixing the acidic tris(2H-hexafluoroisopropyl) borate (THFPB) with Mg(BH_4_)_2_ to study Mg-S batteries) [90]. Note that in the latter case, reaction of the reductive borohydride with the acidic additive was indicated thereby partially transforming the borohydride (further investigation is needed to identify the new species). Another approach that was applied to modify Mg(TFSI)_2_ solutions with BH_4_^−^ was using a BH_4_^−^ anion with hydride groups that were partially substituted with phenol [91]. Both 0.5 M Mg(TFSI)_2_ and 0.15 M Mg(B(OPh)_3_H)_2_ were combined in diglyme and exhibited low deposition overpotentials (*ca* −0.64 V); however, the coulombic efficiency remained at low as 64% even after extended cycling [91].

The successes with utilizing magnesium borohydride in Mg batteries later inspired examining borohydrides electrolytes for Ca batteries, wherein Ca metal anode is notorious for being easily passivated by common electrolytes. Ca(BH_4_)_2_ in tetrahydrofuran (THF) electrolyte was remarkably capable of plating/stripping Ca metal with high coulombic efficiencies, high purity and current densities (Figure 6) [92].

### 4.2. Electrolytes with Wide Electrochemical Window: Closo-Borate Salts 

The stability against electrochemical oxidation of borohydrides in ethereal solutions was found low in all magnesium borohydride based electrolytes (i.e., 1.7, 2.2 and 2.3 V vs. Mg/Mg^2+^) on platinum, stainless steel and glassy carbon electrodes, respectively) [79]. This, coupled with the highly nucleophilic nature, limit the use of borohydrides electrolytes under a high voltage environment and with cathodes susceptible to nucleophilic attacks such as S or some oxides. To increase the oxidative stability of the Mg(BH_4_)_2_ electrolytes, and minimize their hydridic reactive character, strengthening of the B-H bond through forming 3-dimensional B-B bonds as in icosahedral boron clusters (hydroborates and carborates) was proposed and demonstrated using *closo*-carborates [9,93,94]. The concept was that these clusters would retain the high compatibility of the B-H based salt with Mg metal, whilst expanding the electrochemical stability window. Computational and experimental support of this concept was first presented using the dicarborate anion, where carboranyl magnesium chloride electrolyte (1-(1,7-C_2_B_10_H_11_)MgCl) was reported with high oxidative stability (3.3 V vs. Mg) (Figure 7). The electrolyte also demonstrated the high compatibility of the carborate anion with Mg metal (i.e., columbic efficiency was *ca* 100%). Based on this proof of concept, a new simple salt based on the more weakly coordinating monocarborate anion CB_11_H_12_^−^ was synthesized and reported with high solubility in triglyme and tetraglyme (>1M) [94]. The electrolyte efficiently cycled Mg metal (>98%) with very low overpotentials, had high chemical stability and was non corrosive. In addition, unlike the case in the borohydride electrolytes, the Mg^2+^ in this electrolyte was found be unassociated with the anion as revealed in the crystal structure of the isolated salt (Figure 7). It is worth mentioning that the oxidative stability of Mg(CB_11_H_12_)_2_ (measured in acetonitrile solvent), was found to be very high (4.9 V vs. Mg/Mg^2+^) which exceeds that of all ether solvents. These remarkable properties qualified the report of this electrolyte as a breakthrough in the field of Mg electrolytes [2]. A follow up study reported the retention of Mg metal compatibility after fluorination of the C-H bond in the CB_11_H_12_^−^ anion, as was apparent from high coulombic efficiencies (96%) and low overpotentials (measured in 5 mM salt concentrations in triglyme) [95].

### 4.3. Hydride Interfaces in Multivalent Batteries: Significance and Nature

It has been generally accepted that the formation of interfaces on the surface of multivalent metal anodes such as Mg would lead to passivation phenomena, thereby prohibiting the function of the battery. This is interestingly in stark contrast with Li or Na based batteries, where these interfaces or solid electrolyte interfaces (SEIs) (resulting from decomposition of the electrolyte), are permeable to Na^+^, Li^+^ and are instrumental in enabling a stable performance of the anode [97,98]. The root causes of these passivating phenomena are not well understood and could likely result from the divalent nature of these ions which results in low diffusivity in the solid state. 

Recently, research into the function of hydrides electrolytes in Mg and Ca batteries revealed the presence of Mg and Ca interfaces that could support highly efficient deposition/stripping of these multivalent metals. This indeed alters the notion of what has been widely understood about factors required to support the function of these anodes [2]. The first evidence of Mg^2+^ permeable SEI in Mg batteries was shown in the borohydride solutions [99]. The properties of the interface were demonstrated to play an important role in the performance of the electrolyte. For example, the presence of the SEI was observed in Mg/Mg symmetric cells and was found to be dependent on the type of solvent and borohydride additive used, i.e., the use of monoglyme solvent and LiBH_4_ additive (in the 1:3 molar Mg(BH_4_)_2_:LiBH_4_/DME electrolyte) resulted in the lowest cell overpotentials [99]. Therefore detailed investigations of the nature of this interface and morphology were conducted using operando electrochemical-synchrotron soft X-ray absorption (sXAS) and transmission electron microscopy (TEM) [99]. Chemical transformation of the BH_4_^−^, accompanied by H_2_ gas release, into B_12_H_12_^2−^, was observed to occur during the Mg deposition process (Figure 8). It is worth noting that the formation of Mg rich, Li poor alloy was observed in the 1:6 molar Mg(BH_4_)_2_:LiBH_4_/diglyme electrolyte and was suggested to play a role in the Mg deposition process [100]. However, the formation of this alloy seems to be limited thus far to this LiBH_4_ rich mixture as Li presence was not found in the 1:3 molar Mg(BH_4_)_2_:LiBH_4_/DME electrolyte [99]. SEI presence was also revealed in Mg(CB_11_H_12_)_2_ electrolytes and constituted of Mg, B and C species, in addition to a crystalline phase indexed to be close to a MgB_2_O_5_ like species [101,102]. The deposited magnesium was found to be in the form of Mg nanoparticles (*ca* 10 nm) embedded in an amorphous matrix of SEI material whose growth was captured. This morphology and resultant SEI enabled cycling Mg/Mg symmetric cells under unprecedented high current rate conditions (10 mA cm^−2^) [102]. 

In Ca batteries, the presence of CaH_2_ on the surface of Ca metal, formed from the reaction between the THF solvent and Ca metal was revealed [92]. The non-reactivity of Ca(BH_4_)_2_ with the hydride could be the source of the high compatibility in this case. No detailed investigation of possible SEI formation under high currents were reported and further understanding of factors that govern these performances is desired. 

### 4.4. Solid-State Electrolytes

The divalent positive charge carried by cations such as Mg^2+^ leads to sluggish diffusion kinetics in solid-state structures. As such, designing electrolytes with acceptable Mg^2+^ conductivity has been very challenging. However, the report of highly Mg compatible Mg(BH_4_)_2_-based materials as liquid electrolytes, stimulated research that investigated Mg borohydride-based salts as solid electrolytes. This was first demonstrated for Mg(BH_2_)(NH_2_), selected due to the presence of cavities large enough to enable magnesium ion conduction through the hopping mechanism. Conductivity of about 10^−3^ mS/cm was measured at 423 K for Mg(BH_4_)(NH_2_), which is three orders of magnitude higher than that in Mg(BH_4_)_2_, presumably due to the shorter distance between the two nearest Mg atoms (3.59 Å in Mg(BH_4_)(NH_2_) vs. 4.32 Å in Mg(BH_4_)_2_) (Figure 9) [103]. 

Higher conductivities at lower temperatures were reported for *cis*-Mg(en)(BH_4_)_2_, en = NH_2_(CH_2_)_2_NH_2_ [104]. The material was obtained by the mechanochemical processing of the ethylene diamine and Mg(BH_4_)_2_ powder mixture. The mobility of Mg^2+^ ions in *cis*-Mg(en)(BH_4_)_2_ increased from 5 × 10^−8^ S/cm to 6 x 10^−5^ S/cm in the temperature range 303–343 K (Figure 9). 

Polymer-electrolyte systems typically based on polyethylene oxide (PEO) were also reported for Mg batteries. The nanocomposite polymer electrolyte based on PEO, Mg(BH_4_)_2_ and MgO, allowed for reversible Mg deposition/dissolution with 98% coulombic efficiency, however the conductivity was not reported [105]. 

## 5. Hydrides’ Potential as Stable Solvents: Ionic Liquid Electrolytes

To eliminate hazards associated with the use of volatile and flammable battery solvents, ionic liquids, which are salts in the liquid state at room temperature, represent an alternative option [106]. Ionic liquids can have high thermal stabilities and appreciable conductivities which make them attractive for use as solvents in batteries. The most successfully used ionic liquids are those for Li and Na batteries and couple an organic cation (ammonium, phosphonium or pyrrolidinium cations) with a fluorinated anion such as bis(fluorosulfonyl)imide ((FSO_2_)_2_N^−^, FSI) or bis(trifluoromethanosulfonyl) imide TFSI^−^ ((CF_3_SO_2_)_2_N^−^) [106]. However, one of the key challenges in designing ionic liquids lies in the stability of the electrolyte itself or both the electrolyte/anode and the electrolyte/cathode interphases. For instance, an anion with insufficient oxidative stability can limit the choice of cathode material. Likewise, anions susceptible to decompose under the highly reductive anode environment can partially or fully passive the anode. This has been particularly true in magnesium batteries, where fluorinated anions such as TFSI^−^ passivate the Mg metal surface thereby prohibiting the use of ionic liquid solvents (unless highly reducing salts/reagents are added) [6]. Recently, as discussed in Section 4, Mg deposition/stripping was shown possible in electrolytes based on Mg(BH_4_)_2_ salt and TFSI^−^ based ionic liquids [88,89]. Reported systems consisted of a pyrrolidinium cation with dangling PEGylated chains [89] or with ether-functionalized ammonium cation N_2(20201)3_^+^ [88]. The incorporation of chelating ethereal moieties was inspired by previous studies of enhanced Mg(BH_4_)_2_ dissociation and performance in ethers with increased number of oxygen electron donors (Section 4) [79,80]. For example, ionic liquids based on methyl polyethylene glycol MPEG_3_PyrTFSI (three ether oxygens) and MPEG_7_PyrTFSI were superior to *N*-butyl-*N*-methylpyrrolidinium BMPyrTFSI [89]. Further studies showed that ionic liquid solutions with small concentrations of Mg(BH_4_)_2_ (i.e.,18 mM) in 0.3 M Mg(TFSI)_2_/tetraglyme, *N*-methyl-*N*-butyl pyrrolidinium TFSI (1:2 molar) allowed for some reversible Mg deposition/plating (initial coulombic efficiency *ca* 87% that degraded with cycling). Interestingly, despite the presence of Mg(BH_4_)_2_, the anodic stability was reported to be up to 3 V, however it dropped to *ca* 2.1 V in cycled solutions [107].

In the aforementioned studies, given the use of borohydrides, the true oxidative stabilities of these solutions were low (<2.5 V). Therefore, to eliminate challenges associated with the use of strongly reducing agents, a new recent alternate approach designed, for the first time, ionic liquids based on *closo-*borates for Mg batteries [108]. This was inspired from the high compatibility of the carborate anion as was revealed from the Mg salts analogues (Section 4). In addition, the high oxidative stability and weakly coordinating nature of these anion makes them highly desirable for Li and Na batteries. Consideration of ionic liquids based on *closo*-borates in energy storage devices in general has been ignored given the very high melting temperature of these salts attributed to the rigid structure of the anion. However, recently, this challenge was overcome by creating ionic liquids which incorporate the highly stable *closo-*monocarborate anion with ammonium cations decorated with flexible alkoxy ligands (N_2(20201)3_^+^ and N_4(20201)3_^+^), which compensated for the rigidity of the anion. These ionic liquids were reported to remain in the liquid phase down to −52 °C, had high conductivity (in the order of 10^−4^ S/cm) and were highly dissociated (Figure 10) [108]. This placed them amongst best performing ionic liquids being considered for batteries. Importantly, they were used to demonstrate reversible Li and Mg deposition/stripping thereby marking the introduction of new competent ionic liquids family to energy storage devices [108]. 

## 6. Demonstrations of Hydride Electrolytes in Batteries 

Successful demonstrations of batteries require electrolytes that are not only highly conductive, but that are also highly compatible with both the anode and the cathode. Compatibility means: (1) the electrolyte is stable chemically and electrochemically upon contact with the electrode’s components (this includes active materials, other components such as conductive additives, etc.). The consequence of this is the absence of any decomposition products from the electrolyte on the surface of the electrode and (2) the electrolyte is unstable in the environment of the electrode and decomposes generating a surface film on the electrode. These decomposition products however, allow for the passage of the charge carriers (i.e., cations) and importantly serve as a barrier to prevent further decomposition of the electrolyte. The surface layers formed on the anode side are referred to as SEI (Solid Electrolyte Interface) as discussed in Section 4 and those formed on the cathode are called CEIs (Cathode Electrolyte Interfaces). The electrolyte–electrode interactions are complex in nature and highly dependent on the electrode and its components. 

The first example of the successful application of hydrides electrolytes in a stably cycling battery cell was reported in a Mg battery for Mg(BH_4_)_2_–LiBH_4_/DME electrolyte with a magnesium anode and a Chevrel phase Mo_6_S_8_ cathode (Figure 5). The cell was operated at room temperature and delivered the expected performance from the Mo_6_S_8_ cathode [79]. A follow-up study using a similar battery (rate of C/10), delivered a capacity of 99 mAh/g during the initial discharge (vs. 129 mAh/g of the theoretical value) and then retained 90% of this capacity for over 300 cycles [80]. 

LiBH_4_ application as a solid-state electrolyte in all solid-state batteries was first reported in an electrochemical cell with Li metal anode and LiCoO_2_ cathode [109]. The battery performance was evaluated at 393 K, which ensured the stability of the highly conductive LiBH_4_ phase. Due to the nucleophilic nature of borohydrides, parasitic reactions between LiBH_4_ and LiCoO_2_ occurred leading to high interfacial resistance and significant capacity loss. To mitigate this interaction, amorphous Li_2_PO_4_ was introduced between electrolyte and the cathode and somewhat improved the battery cycling performance; i.e., 97% of the initial discharge capacity (89 mAh/g) after 30 cycles. The hydritic nature of the borohydride called for investigating other cathodes wherein stable interface formation between LiBH_4_ and TiS_2_ cathode was reported [41,110]. The solid-state battery consisted of TiS_2_ cathode, LiBH_4_ electrolyte and Li anode and was successfully cycled 300 times at 393 K and a rate of 0.2 C. Although, the initial discharge capacity was only 80 mAh/g (vs. 239 mAh/g of the theoretical value), at the second cycle the parameter reached 205 mAh/g with close to 100% coulombic efficiency. The cell’s durability was explained by the chemical/electrochemical oxidation of the borohydride just below the TiS_2_ surface. This was accompanied by hydrogen release and formation of a new phase, identified to most likely be Li_2_B_12_H_12_. Later studies examined this cathode with other LiBH_4_ based electrolytes such as Li_4_(BH_4_)_3_I, combined with either TiS_2_/LiBH_4_ or TiS_2_/Li_4_(BH_4_)_3_I composite electrodes [111]. The higher durability of the cell consisting of TiS_2_/Li_4_(BH_4_)_3_I composite electrodes underscored the importance of the high mechanical plasticity in the I^−^-substituted LiBH_4_. 

The feasibility of lithium-sulfur (Li-S) conversion batteries utilizing a borohydride electrolyte was demonstrated at 393 K [112]. The Li-S cell delivered 1140 mAh/g at 0.05 C rate, during the first discharge, which corresponded to 70% of sulfur utilization ratio. Over the next 45 cycles, the discharge capacity remained as high as 730 mAh/g with nearly 100% coulombic efficiency. Further studies using the more Li^+^ conductive electrolytes such as LiBH_4_–nanoconfined LiBH_4_ [113] LiCl [114] or allowed for lowering the operating temperature of the Li-S cell. The surprising performances of these batteries, despite the highly reactive nature of borohydrides towards S, suggest formation of a CEI (Cathode Electrolyte Interface) that is analogous to that observed on TiS_2_; i.e., *closo*-borates. In fact, just recently, excellent Li/S battery performance was reported in a 7:3 molar [CB_9_H_10_^−^:CB_11_H_12_]^2−^ Li salt [70].

The low oxidative stability of the borohydride electrolytes limited the demonstration of high voltage batteries. This was particularly apparent in liquid borohydride electrolytes used in Mg batteries, which unlike solid-state electrolytes, lacked the presence of the kinetic barriers that could extend the oxidative decomposition onset. However, the high stability of Mg(CB_11_H_11_)_2_ in tetraglyme allowed for cycling the high voltage cathode α-MnO_2_ (initial discharge capacity of 170 mAh/g, Figure 7). This was the first successful example of a Mg coin cell operated at voltages close to 3 V vs. Mg^2+^/Mg (i.e.,4.2 V vs. Li^+^/Li) (Figure 7) [94]. On the other hand, one of the best demonstrations of a high voltage solid-state battery (3V), was for the Na metal battery employing Na_2_(B_12_H_12_)_0.5_(B_10_H_10_)_0.5_ electrolyte which was operated for 250 cycles [115]. One key to this successful demonstration was the high oxidative stability of the electrolyte and accomplishing a good contact between the cathode (NaCrO_2_, theoretical capacity of 120 mAh/g) and the electrolyte (this was achieved by dissolving the latter in anhydrous methanol and dispersing NaCrO_2_ into the solution, followed by drying in vacuum and heat treatment at 543 K in order to recrystallize the electrolyte). A follow-up study utilized isopropanol in order to recrystallize the electrolyte in the cathode at a lower temperature (453 K) [116].

Hydride electrolytes were also examined in batteries employing insertion and alloy anodes in liquid Mg batteries (i.e., good cycling was reported for Bi anode in a borohydride electrolyte [117]), solid-state lithium ion (i.e., nanostructured Bi_2_Te_3_/LiBH_4_ composite [118] or Si with Li_2_B_12_H_12_ [18]) and sodium ion batteries (Na_3_Sn anode in Na(B_12_H_12_)_0.5_(B_10_H_10_)_0.5_ electrolyte) [116]. It is noted that performance losses due to challenges associated with anodes such as Sn and Si occurred and made it difficult to establish the potential advantage of hydrides in these batteries. 

## 7. Conclusions

Several decades ago, hydrides were widely investigated as hydrogen storage materials and their potential to electrochemical energy storage devices was limited to the Ni-MH batteries. However, just over a decade ago, this has been altered owing to two key discoveries: First is the discovery of high solid-state ionic conductivity in LiBH_4_, which opened the doors for designing hydrides as highly conducting solid-state electrolytes for Li and Na batteries. The second was owing to the creation of borohydrides electrolytes that are highly compatible with Mg metal. This opened a new design space in Mg electrolytes as it resulted in highly competent boron hydrogen-based electrolytes which overcome key challenges and changed several preconceptions about what an ideal electrolyte needs to be like. These advancements have also served to inspire investigating several new families of electrolytes for Mg batteries.

The high Li^+^ and Na^+^ conductivities in *closo*-borate solid-state electrolytes allowed for battery demonstrations and have placed them amongst the top performing solid-state electrolytes being investigated. On the other hand, *closo*-carborate liquid electrolytes have enabled the demonstrations of high voltage Mg batteries for the first time in battery coin cells. 

The journey of hydrides in the field of energy dense batteries is just beginning. Studies of key factors that govern the performances of these materials would be advantageous for future developments. In addition, the high costs associated with *closo*-borates could represent a barrier towards their practical use and therefore overcoming this challenge would drive the consideration of these materials as future battery electrolytes. 

## Figures and Tables

**Figure 1 molecules-25-01791-f001:**
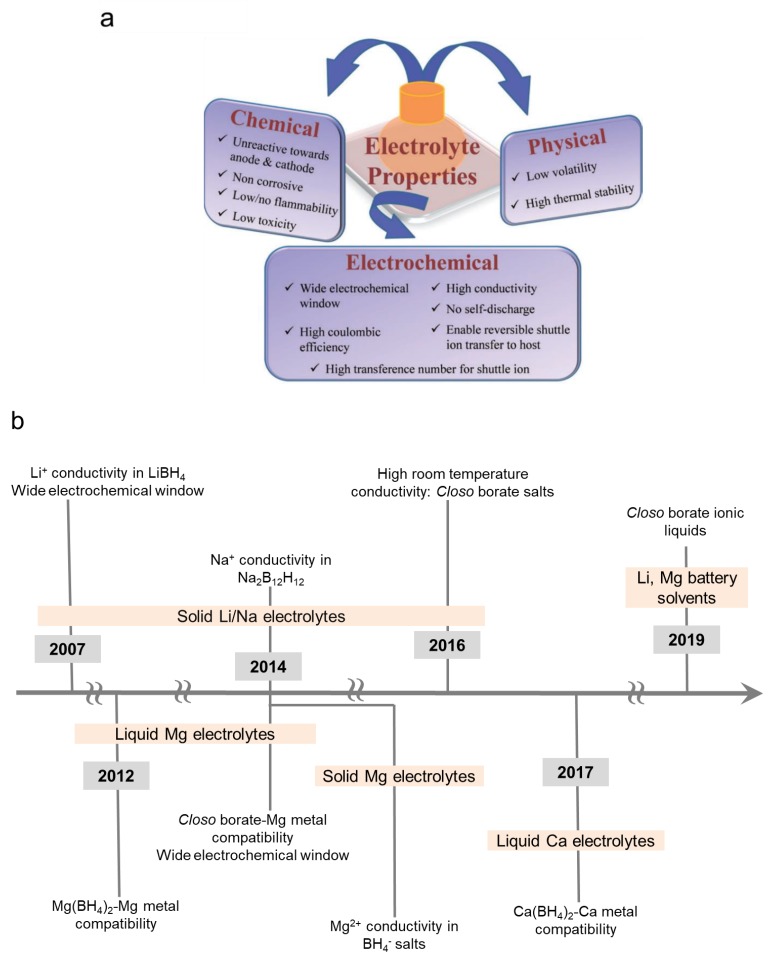
(**a**) Key properties required for competent battery electrolytes. Reprinted with permission from reference [9], copyright 2014 WILEY-VCH. (**b**) Timeline depicting key advancements made in the development of hydrides as battery electrolytes.

**Figure 2 molecules-25-01791-f002:**
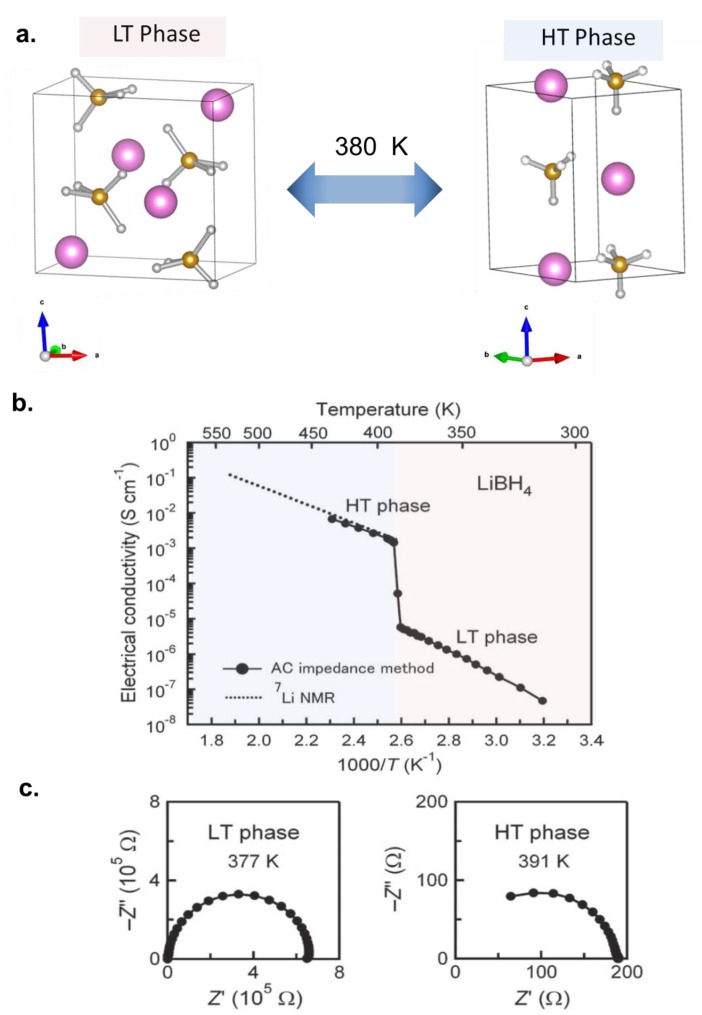
Properties of solid-state electrolytes for Li batteries: (**a**) Phase transition in LiBH_4_, (**b**) Arrhenius plot depicting the ionic conductivity of LiBH_4_, (**c**) Impedance spectra of the low and high temperature LiBH_4_ phases. Panels **b** and **c** are reproduced with permission from reference [11], American Institute of Physics.

**Figure 3 molecules-25-01791-f003:**
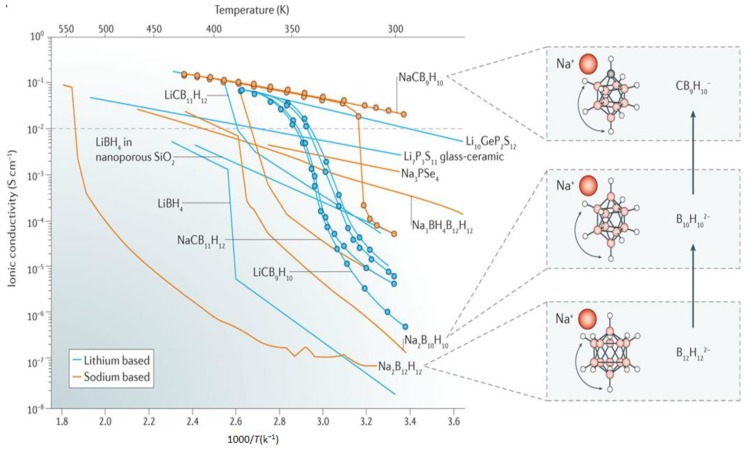
Arrhenius plots depicting the cationic conductivity in a variety of hydrides. Reprinted with permission from reference. [8], copyright 2016 Macmillan Publishers Limited, part of Springer Nature.

**Figure 4 molecules-25-01791-f004:**
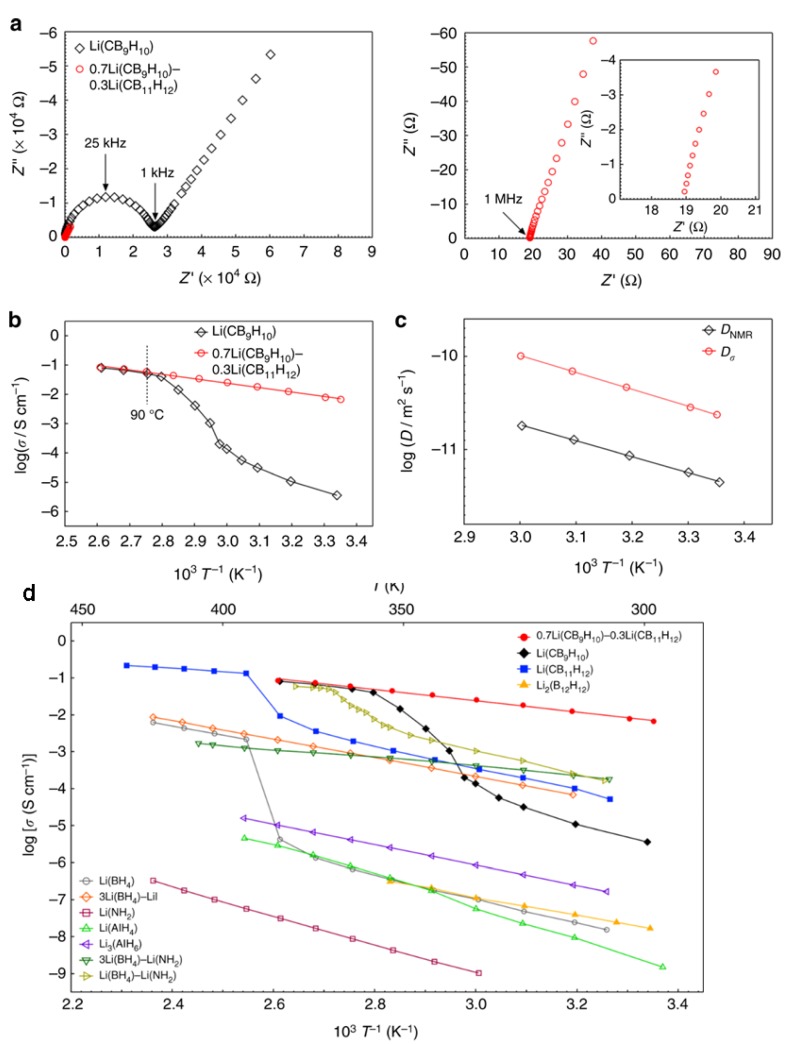
Lithium ion conductivity of 7:3 molar (CB_9_H_10_:CB_11_H_12_)^2−^ Li salt. (**a**) Nyquist plots at room temperature and magnified Nyquist plots, (**b**) Arrhenius plots of the lithium ion conductivity, (**c**) Arrhenius plots of the diffusion coefficients calculated from the impedance and nuclear magnetic resonance (NMR) measurements, (**d**) Arrhenius plots comparing the conductivity for variety of Li salts. Reprinted with permission from reference [70], copyright 2019 Macmillan Publishers Limited, part of Springer Nature.

**Figure 5 molecules-25-01791-f005:**
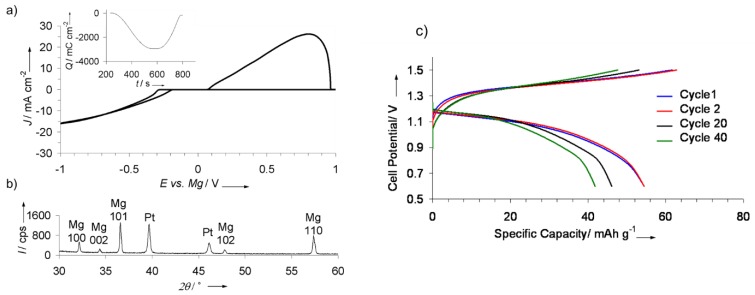
Performance of Mg borohydride electrolyte: (**a**) Cyclic voltammogram (inset shows deposition/stripping charge balance) in 1:3 molar Mg(BH_4_)_2_:LiBH_4_ in 1,2 dimethoxyethane (DME), (**b**) XRD results following galvanostatic deposition of Mg metal, (**c**) Battery cycling with this electrolyte. Reprinted from reference [79] with permission. Copyright © 2012, Wiley-VCH Verlag GmbH and Co. KGaA, Weinheim.

**Figure 6 molecules-25-01791-f006:**
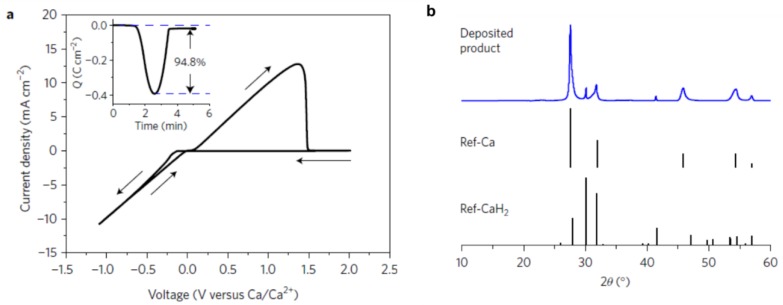
Ca(BH_4_)_2_/THF electrolyte for Ca battery: (**a**) Electrochemical results of calcium plating/stripping in 1.5 M Ca(BH_4_)_2_/THF, (**b**) Powder X-ray diffraction demonstrating Ca plating from the electrolyte. Reprinted with permission from reference [92], copyright 2017 Macmillan Publishers Limited, part of Springer Nature.

**Figure 7 molecules-25-01791-f007:**
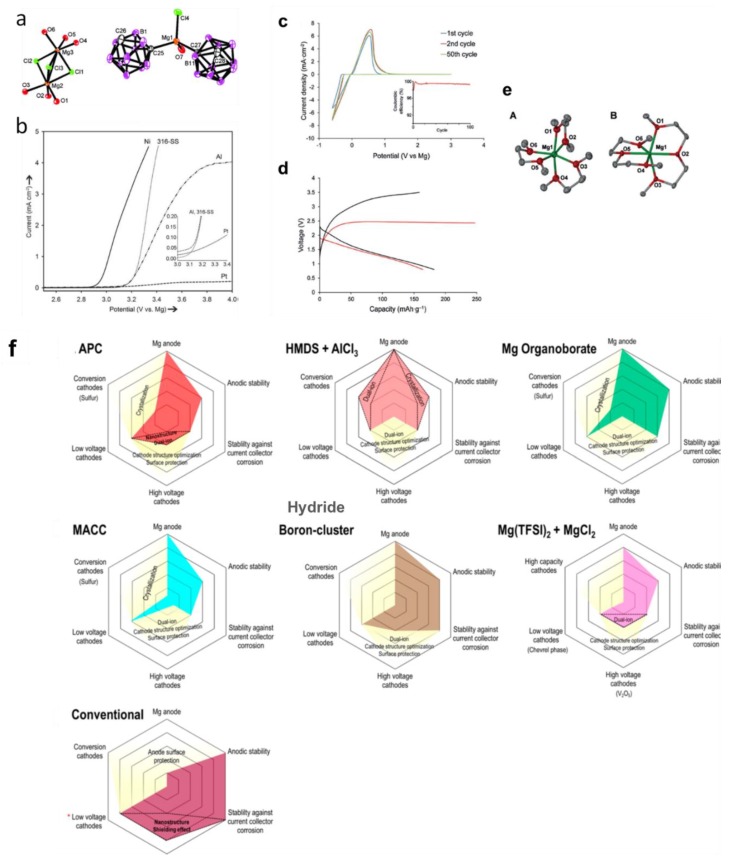
For 1-(1,7-C_2_B_10_H_11_)MgCl electrolyte: (**a**) Molecular structure and (**b**) Oxidative stability. Reprinted from reference [93] with permission from Wiley-VCH Verlag GmbH and Co. KGaA, Weinheim, Copyright 2014. For Mg(CB_11_H_12_)_2_/tetraglyme electrolyte: (**c**) Cyclic voltammograms on Pt, (**d**) Initial discharge–charge profiles of a rechargeable Mg battery with Mg(CB_11_H_12_)_2_/tetraglyme (black line) and chlorophenyl aluminate electrolyte APC (red line) as the electrolyte, a Mg anode, and α-MnO_2_ cathode, (**e**) Mg cation coordination environment in monoglyme (A) and diglyme (B). Reprinted from reference [94] with permission from Wiley-VCH Verlag GmbH and Co. KGaA, Weinheim, Copyright 2015. (**f**) Position of hydride liquid electrolytes (boron clusters) amongst other Mg battery electrolytes. Reprinted from reference [96] with permission from the American Chemical Society, Copyright 2016.

**Figure 8 molecules-25-01791-f008:**
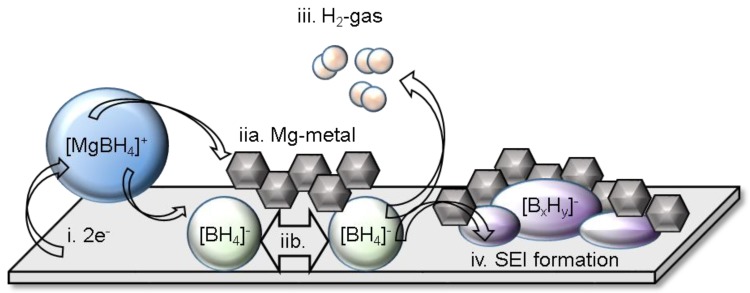
Formation of interface in Mg interface in Mg(BH_4_)_2_:LiBH_4_/DME electrolyte. Reprinted from reference [99], permission of the American Chemical Society, copyright 2017.

**Figure 9 molecules-25-01791-f009:**
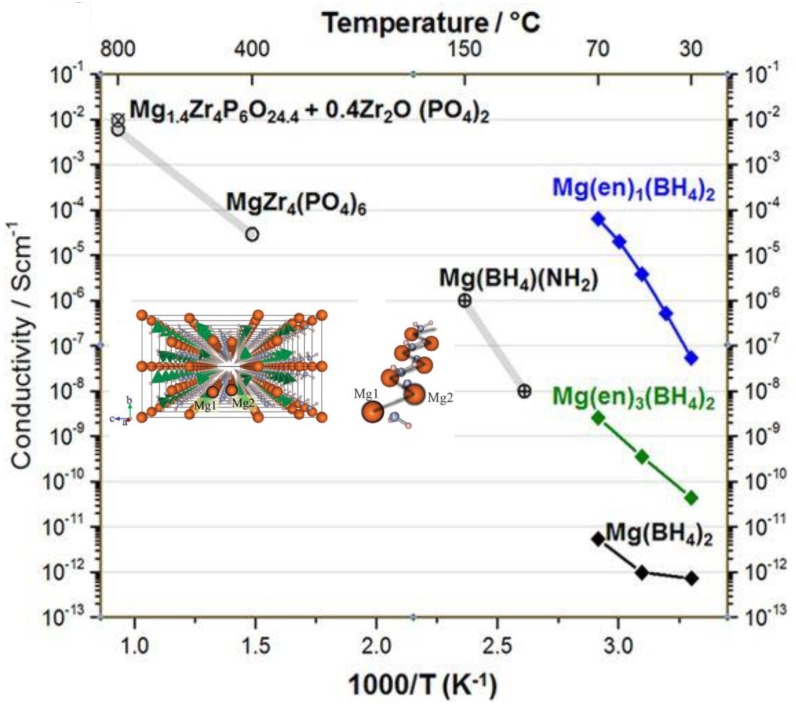
Temperature dependence of ionic conductivity for selected magnesium solid-state electrolytes. The structure of Mg(BH_4_)(NH_2_) is shown. Reproduced from reference [104] permission from Springer Nature, Copyright 2017.

**Figure 10 molecules-25-01791-f010:**
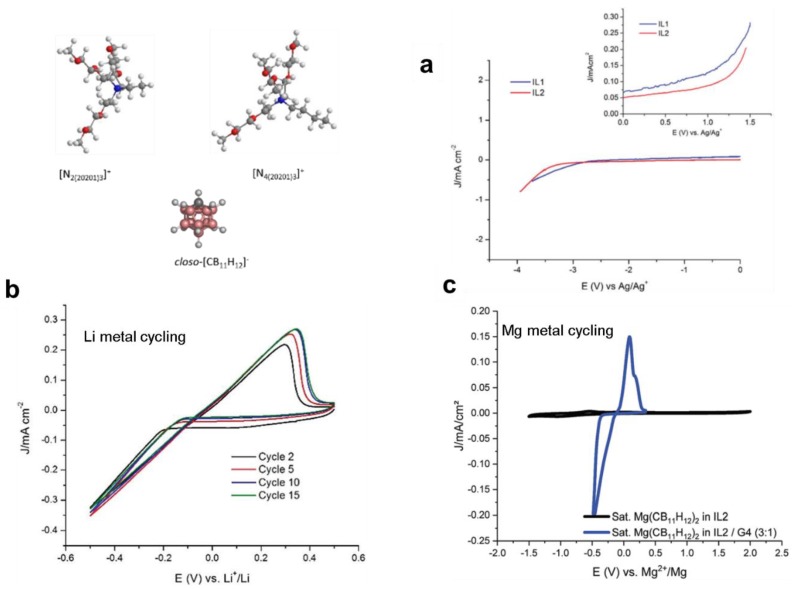
For the *closo*-carborate ionic liquids (structure shown): (**a**) Oxidative and reductive stability, (**b**) and (**c**) are Li and Mg electrochemical deposition/stripping, respectively. Reproduced from reference [108] with permission from the Royal Society of Chemistry, Copyright 2019.

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
