# Peer review of "Beyond Typical Electrolytes for Energy Dense Batteries"

_molecules, 2020, doi:10.3390/molecules25081791_

Round 1

Reviewer 1 Report

Comments to the Author The work by Rana Mohtadi. reviews recent progress in hydride-based electrolyteselectrolyte for energy dense batteries. The authors focused their work on classes of hydride-based electrolytes for both monovalent batteries andmultivalent conduction including recent publications.Nowadays, the hydride-based electrolyte has been paving a path to overcome the flammability/volatility of thepopular lithium ion batteries electrolytes, which is apromising topic and less explored. The manuscript is well written and organized. This review is expected to be of broad interest especially to the battery electrolytecommunity if the following issues are solved properly.

1. Overall, this review lacks theoretical depth to some extent. The author needs to discuss more about the fundamental roles, features, and requirements of hydride based-electrolyte in section 2. 2. The article discussed many types of hydride-basedelectrolyte including LiBH4, Li2(BH4)(NH2), Na2(BH4)(NH2) etc. However, the articles need to discuss more about the fundamental function mechanisms for different kinds of hydride in addition to briefly present the performance of ion conductivity, other parameters to evaluate the electrolyte are also welcome. More insights of the approaches to overcome the challenges that the current hydride electrolyte faced should be provided in the review. 3. The statement ofUnlike the case for the lithium salts, formation of borohydrides that are highly Na+conductive was less successful. 40-42 What is the underlying cause of this difference? Does the reference article give an explanation, the author may give moreintroduction? 4. The statement ofTo further improve the cationic conductivity, the utilization of carborane boronclusters, which carry a monovalent negative charge, was investigated for LiCB11H12 and NaCB11H12. The statement in this article makes it seem that the key roleon improving the conductivity is the monovalentnegative charge. Please explain why? Or if there are statements warranted by the citation, no further explanation but statement can be cited here. 5. The statement “The properties of the interface weredemonstrated to play an important role in the performance of the electrolyte. For example, the presence of the SEI was observed in Mg/Mg symmetric cells and was found to be dependent on the type of solvent and borohydride additive used” What type of solvent and additives did this reference used? More insight about the solvent and additive selectionneed to be discussed. 6. The statement “It is worth noting that formation of Mg rich, Li poor alloy was observed in the 1:6 molar Mg(BH4)2:LiBH4/diglyme electrolyte and was suggested to play a role in the Mg deposition process” what is the role and does the literature mention how the molar ratio affects formation the this kind of alloy? 7. Last but not least, the article is very difficult to read and not easily comprehensible. The article uses a lot of space to introduce similar examples, in my private opinion, it is sufficient to elaborate a theory and viewpoint method with one classic case. Meanwhile, some concise diagrams will be very useful to help readers better understand the article and attract more interest. Therefore, I suggest its consideration for publication at molecules after major revision. 

Author Response

“The work by Rana Mohtadi. reviews recent progress in hydride-based electrolyteselectrolyte for energy dense batteries. The authors focused their work on classes of hydride-based electrolytes for both monovalent batteries andmultivalent conduction including recent publications.Nowadays, the hydride-based electrolyte has been paving a path to overcome the flammability/volatility of thepopular lithium ion batteries electrolytes, which is apromising topic and less explored. The manuscript is well written and organized”

Ans. I thank the reviewer for the positive comments

“1. Overall, this review lacks theoretical depth to some extent. The author needs to discuss more about the fundamental roles, features, and requirements of hydride based-electrolyte in section 2. 2. The article discussed many types of hydride-basedelectrolyte including LiBH4, Li2(BH4)(NH2), Na2(BH4)(NH2) etc. However, the articles need to discuss more about the fundamental function mechanisms for different kinds of hydride in addition to briefly present the performance of ion conductivity, other parameters to evaluate the electrolyte are also welcome. More insights of the approaches to overcome the challenges that the current hydride electrolyte faced should be provided in the review.”

Ans. In the review, I cover key hydride fundamental properties; i.e. structural, chemical.. that were found to be central in enabling the electrochemical function of these electrolytes. This is clearly presented throughout the manuscript; i,e. in reference to the salts brought up by the reviewer LiBH4, Li2(BH4)(NH2), see ln 100-103, ln 133-142. The purpose of this review is to cover the breadth of hydride applications as electrolytes, therefore other very specific material details would be a topic for a separate specialized review. In addition, note that section 2 is merely an introductory section that lays out the review theme and topics covered in the next sections, therefore more information is not needed. To avoid confusion about its function, I added this information in Ln 79-85.

“3. The statement of “Unlike the case for the lithium salts, formation of borohydrides that are highly Na+conductive was less successful. 40-42” What is the underlying cause of this difference? Does the reference article give an explanation, the author may give moreintroduction?”

Ans. This statement is my own insight to the field that Na borohydride salts underperforms in terms of the cationic mobility. I hope it will drive interests in fundamental and computational studies to determine the root causes.

“4. The statement of “To further improve the cationic conductivity, the utilization of carborane boronclusters, which carry a monovalent negative charge, was investigated for LiCB11H12 and NaCB11H12.” The statement in this article makes it seem that the key roleon improving the conductivity is the monovalentnegative charge. Please explain why? Or if there are statements warranted by the citation, no further explanation but statement can be cited here”

Ans. The reviewer seems to have missed the information right in the same paragraph where I clearly explain the function of the carborane. As I introduce the anions, I merely mention that they have a negative charge, which I explain later that it also plays a role. This is how the paragraph reads:     ” To further improve the cationic conductivity, the utilization of carborane boron clusters, which carry a mononvalent negative charge,  was investigated for LiCB11H12 and NaCB11H12.67 Interestingly, LiCB11H12 and NaCB11H12 underwent transition to a superconducting (> 0.1 S/cm) disordered phase at much lower temperatures, 400 K and 380 K respectively (Fig.3), accompanied with high rate of anion reorientational jumps (1010–1011 jumps/ s). The cluster’s monovalent charge, presence of less neighbors (cation:anion molar ratio = 1:1), and increased lattice constant were hypothesized to cause these enhanced conductivities.  These were confirmed from ab initio molecular dynamics (AIMD) which also demonstrated that formation of a dipole (carbon atoms), creates a frustrated lattice and counteracts the ability of the phase to order, thereby reducing the transition temperature to superconducting phases”

“5. The statement “The properties of the interface weredemonstrated to play an important role in the performance of the electrolyte. For example, the presence of the SEI was observed in Mg/Mg symmetric cells and was found to be dependent on the type of solvent and borohydride additive used” What type of solvent and additives did this reference used? More insight about the solvent and additive selectionneed to be discussed”

Ans. This information has been added (ln 513-514).

The statement “It is worth noting that formation of Mg rich, Li poor alloy was observed in the 1:6 molar Mg(BH4)2:LiBH4/diglyme electrolyte andwas suggested to play a role in the Mg deposition process” what is the role and does the literature mention how the molar ratio affects formation the this kind of alloy?

Ans. There are no such studies and the author would prefer not to speculate given the absence of literature information.

Last but not least, the article is very difficult to read and not easily comprehensible. The article uses a lot of space to introduce similar examples, in my private opinion, it is sufficient to elaborate a theory and viewpoint method with one classic case. Meanwhile, some concise diagrams will be very useful to help readers better understand the article and attract more interest.

Ans. This review covers a wide range of topics in order to capture critical developments of the field of hydrides as electrolytes in diverse battery technologies. So any difficulty the reviewer may have encountered is due to the wide range of exciting developments in this field. Whilst I am not expecting one reader to be an expert with all the topics covered,  I aim to bring these topics to the attention of the broader readership to further investigate and learn more about these topics. Regarding the diagrams and figures, I respectively disagree with the reviewer. The review includes many Fig. /diagrams that are concise where they summarize a wide variety of materials or concepts, i.e. see Fig.1, Fig.2, Fig.3, Fig.4, Fig. 5… ..diagrams such as Fig. 1, Fig 8. In addition, I added Fig,7f to show hydrides position as liquid Mg electrolytes.

Reviewer 2 Report

This article is exceptional extensive: About 9000 words (exclusive references), 120 references, 10 complex figures (with several parts). To read it is an complex exercise: the nature of the text varies from very detailed and technical to very general and speculative. The structure of the paper is not explained. In this form the paper should not published.

The paper pretends to be a review and in some way it does be a review, considering many publications and trying to judge the state of the art with regard to the development of new batteries. But this review is presented in a narrative way not in a structured way. In this narrative discourse no division is made between details and specific complications and highlights and general characteristics and limitations. With help of a predefined structural approach grasp will be offered to readers in the multitude of relevant material.

The subject of the article is surely interesting and I would applaud when the author would revise the material to a much shorter, well structured paper. I offer some advise.

Make first clear what the problem is with the present energy-dense battery,  like Li-ion systems and discuss this in systematic way. The paper mentions the properties of the electrolyte (flammability, volatility, stability), the nature of anodic and cathodic materials (monovalent multivalent), issues about weight and costs, efficiency of stacking, (dis)charging rate, among others. Most of the readers are aware of some of these issues, but almost nobody has a full overview. To assess the potential and the limitations of new approaches – as discussed in the paper – this overview as a starting point is utmost important. In section 2 a part of such structure is given in figure 1, but all issues should be addressed systematically a forehand of the discussion of new approaches. Use this overview to derive some quantitative (e.g. mAh/cm3) en qualitative parameters that could be used to measure progress in performance of new considered materials and systems. Use this scheme also to define strategies to improve the performance. Order the material in section 3 – 5 in line with these strategies. Each specific approach should not be discussed in all technical details but in line with the strategy with attention to potential drawbacks and limitations. Assess after the evaluation of specific approaches integrally to arrive at the concluding lines se section 7. Be generally aware that the discussion in the article is focused on a subset of the batteries: the light weight high power many cycle segment. For stationary low cost battery systems other developments could be expected.

Author Response

“This article is exceptional extensive: About 9000 words (exclusive references), 120 references, 10 complex figures (with several parts). To read it is an complex exercise: the nature of the text varies from very detailed and technical to very general and speculative. The structure of the paper is not explained. In this form the paper should not published. The paper pretends to be a review and in some way it does be a review, considering many publications and trying to judge the state of the art with regard to the development of new batteries. But this review is presented in a narrative way not in a structured way. In this narrative discourse no division is made between details and specific complications and highlights and general characteristics and limitations. With help of a predefined structural approach grasp will be offered to readers in the multitude of relevant material”

Ans. The reviewer’s comments describing the review are not only untrue but are also bizarre and even contradictory, i.e. “The paper pretends to be a review and in some way it does be a review”. I am afraid that the reviewer seems to have missed what is presented in this review. The review is written by a trend setter in the field of hydrides who is accredited with critical technology advancing accomplishments. It is a culmination of the writer’s deep knowledge and experiences in the field.

In addition, this aim of this review is to cover the breadth of hydride applications as electrolytes, so any difficulty the reviewer may have encountered is may be due to the wide range of exciting developments in this field. Whilst I am not expecting one reader to be an expert in all the topics covered, I aim to bring these topics to the attention of the broader readership to further investigate and learn about these advancements.

 “Make first clear what the problem is with the present energy-dense battery,  like Li-ion systems and discuss this in systematic way. The paper mentions the properties of the electrolyte (flammability, volatility, stability), the nature of anodic and cathodic materials (monovalent multivalent), issues about weight and costs, efficiency of stacking, (dis)charging rate, among others. Most of the readers are aware of some of these issues, but almost nobody has a full overview.” 

Ans. This general information is already presented as appropriate in the introduction (Ln 23-51) and in the sections (i.e. Section 4). The reader is directed to the ref. for further details as they are beyond the scope of this article.

In section 2 a part of such structure is given in figure 1, but all issues should be addressed systematically a forehand of the discussion of new approaches. Use this overview to derive some quantitative (e.g. mAh/cm3) en qualitative parameters that could be used to measure progress in performance of new considered materials and systems. Use this scheme also to define strategies to improve the performance.

Ans. In Fig. 1 and in the related text, I explain key metrics (ln 61-68) relevant to this review. As for the structure of the article, in section 2, I added information that explains the structure of this article.

“Order the material in section 3 – 5 in line with these strategies. Each specific approach should not be discussed in all technical details but in line with the strategy with attention to potential drawbacks and limitations. Assess after the evaluation of specific approaches integrally to arrive at the concluding lines se section 7. Be generally aware that the discussion in the article is focused on a subset of the batteries: the light weight high power many cycle segment. For stationary low cost battery systems other developments could be expected”

Ans. In section 2, I added information that explains the structure of the article.  The discussion in each section already clearly and concisely presents key developments related to a desired property from an electrolyte. i.e. conductivity or compatibility with battery components etc. Therefore, I don’t see the point from the reviewer’s suggestion. In addition, I prefer not to discuss details about applications, i.e. stationary, as this is premature and doesn’t bring useful information (hydride electrolytes are only in the research stage).

Reviewer 3 Report

This review reported the hydride-based electrolytes for energy dense batteries. The authors discussed the developments from two main aspects in detail, the discovery of high Li+ conductivity in the solid state and the high compatibilities of boron-hydrogen salts with Mg metal which are important topics for developing next generation of high energy dense batteries. I think this review is meaningful and can provide some guidance for the future research. Before it is recommended to be published, some modifications should be made to further improve this manuscript. 1. The title of this manuscript should be more specific, “Electrolytes for Energy Dense Batteries” is too broad. 2. Page 3, line 67, “whilst low/non-volatility is highly desired to minimize safety risks”, actually, the low/non-flammability should be quite important to battery safety. 3. Also, in page 1 line 28-34, “For example, in contrast to batteries utilizing liquid electrolytes, the use of solid state electrolyte allows for efficient bipolar stacking design of batteries that decreases the dead space between single cells thereby increasing the overall energy density whilst eliminating the use of volatile liquid electrolytes.” There are several methods to improve the safety of the electrolyte, the design of solid state electrolyte is indeed an important branch for safer battery, the authors are recommended to read the following reference: Wang Q, Jiang L, Yu Y, et al. Progress of enhancing the safety of lithium ion battery from the electrolyte aspect[J]. Nano Energy, 2019, 55: 93-114. 4. The authors are suggested to add a subsection about the introduction of energy dense batteries including the Mg batteries. For example, the characteristics of Mg batteries and their special requirements for electrolytes. 5. For subsections 2: Hydrides as battery electrolytes, this part is too simple, it is recommended to add more introduction and literature research to further enrich the content of this part and help readers to further understand it. 6. The authors are recommended to add figures that summarized the advantages and disadvantages of current electrolytes for energy dense batteries and their contrast to hydride-based electrolytes. I think this can help future researches to better understand the currently problems and the future research directions.

Author Response

“This review reported the hydride-based electrolytes for energy dense batteries. The authors discussed the developments from two main aspects in detail, the discovery of high Li+ conductivity in the solid state and the high compatibilities of boron-hydrogen salts with Mg metal which are important topics for developing next generation of high energy dense batteries. I think this review is meaningful and can provide some guidance for the future research.”

Ans.  I thank the reviewer for the positive feedback

“1.The title of this manuscript should be more specific, “Electrolytes for Energy Dense Batteries” is too broad.”

Ans. I agree that it is broad, however, my aim is to attract the broader battery field to developments of hydrides as electrolytes. Therefore, I prefer to keep it as such.

“2.Page 3, line 67, “whilst low/non-volatility is highly desired to minimize safety risks”, actually, the low/non-flammability should be quite important to battery safety.”

Ans. Current batteries such as commercially available Li ion already employ flammable electrolytes. So, non flammability is not required at this point, however it is rather desired. I would rather keep the sentence as is to avoid any confusions.  

“Also, in page 1 line 28-34, “For example, in contrast to batteries utilizing liquid electrolytes, the use of solid state electrolyte allows for efficient bipolar stacking design of batteries that decreases the dead space between single cells thereby increasing the overall energy density whilst eliminating the use of volatile liquid electrolytes.” There are several methods to improve the safety of the electrolyte, the design of solid state electrolyte is indeed an important branch for safer battery, the authors are recommended to read the following reference: Wang Q, Jiang L, Yu Y, et al. Progress of enhancing the safety of lithium ion battery from the electrolyte aspect[J]. Nano Energy, 2019, 55: 93-114.

Ans. I thank the reviewer for bringing the paper to my attention and I cited it.

“The authors are suggested to add a subsection about the introduction of energy dense batteries including the Mg batteries. For example, the characteristics of Mg batteries and their special requirements for electrolytes”

Ans. In section 4, Ln 352-372, I concisely explained these batteries and their key challenges. For details, the reader can refer to the ref. therein as this is beyond the scope and focus of the article.

“5.        For subsections 2: Hydrides as battery electrolytes, this part is too simple, it is recommended to add more introduction and literature research to further enrich the content of this part and help readers to further understand it”

Ans. Section 2 is merely an introductory section that lays out the review’s theme and topics covered in the next sections. To avoid confusion about its function, I added this information in Ln 79-85.

“6.The authors are recommended to add figures that summarized the advantages and disadvantages of current electrolytes for energy dense batteries and their contrast to hydride-based electrolytes. I think this can help future researches to better understand the currently problems and the future research directions”

Ans. Existing figures cover the position of hydrides per topic, i.e. Fig. 3, Fig. 9. In Fig. 7, I added a Fig.7f  that shows the hydrides’ position as liquid electrolytes in multivalent batteries.

Round 2

Reviewer 1 Report

The authors have done the necessary revisions. It can be accepted now.

Author Response

Thank You for Your consideration.

Reviewer 2 Report

The paper has been slightly adapted. In section two a few sentences are added (marked yellow) offering a overview of the structure. Also other sentences are marked yellow in the new version. However these sentences are not changed. 

Further advises to revise the structure and the line of reasoning are rejected. So the paper has not improved in these aspects.

So I my opinion remains "In this form the paper should not published."

Author Response

1) The title was slightly altered 2) Table of content was added to make the review structure clear 3) Section 2 title was changed to reflect its real purpose, which is the explanation of the review's structure. In addition, I added information in this section that concisely explained the structure of this review and demonstrated electrolyte property-topic relation per comments of reviewer no.2. 4) I added a Fig.1b, which is a timeline of advancements made to make the reader upfront understand the mission and structure of the review. 5) Sections/subsections titles were appropriately adjusted to reflect the electrolyte property being discussed per comments of reviewer no.2.

Reviewer 3 Report

OK.

Author Response

Thank You for Your consideration.